# Spatio-temporal Partial Sensing Forecast of Long-term Traffic

**Zibo Liu**\*                                                             *ziboliu@ufl.edu*
**Zhe Jiang**                                                            *zhe.jiang@ufl.edu*
**Zelin Xu**                                                              *zelin.xu@ufl.edu*
**Tingsong Xiao**                                                      *xiaotingsong@ufl.edu*
**Zhengkun Xiao**                                                          *xiaoz@ufl.edu*
**Yupu Zhang**                                                           *y.zhang1@ufl.edu*
*Department of Computer & Information Science & Engineering, University of Florida*

**Haibo Wang**                                                            *haibo@ieee.org*
*Department of Computer Science, University of Kentucky*

**Shigang Chen**                                                          *sgchen@ufl.edu*
*Department of Computer & Information Science & Engineering, University of Florida*

**Reviewed on OpenReview:** *https://openreview.net/forum?id=Ff08aPjVjD*

## Abstract

Traffic forecasting uses recent measurements by sensors installed at chosen locations to forecast the future road traffic. Existing work either assumes all locations are equipped with sensors or focuses on short-term forecast. This paper studies partial sensing forecast of long-term traffic, assuming sensors are available only at some locations. The problem is challenging due to the unknown data distribution at unsensed locations, the intricate spatio-temporal correlation in long-term forecasting, as well as noise to traffic patterns. We propose a **S**patio-temporal **L**ong-term **P**artial sensing **F**orecast (**SLPF**) model for traffic prediction, with several novel contributions, including a rank-based embedding technique to reduce the impact of noise in data, a spatial transfer matrix to overcome the spatial distribution shift from sensed locations to unsensed locations, and a multi-step training process that utilizes all available data to successively refine the model parameters for better accuracy. Extensive experiments on several real-world traffic datasets demonstrate its superior performance. Our source code is at `https://github.com/zbliu98/SLPF`

## 1 Introduction

**Background:** The *traffic forecast* problem is to use the recent measurements by sensors installed at chosen locations to forecast the future road traffic. This problem has significant practical values in traffic management and route planning, considering the ever worsening traffic conditions and congestion in cities across the world. Most existing work considers a *full-sensing* scenario, where permanent sensors are installed at all locations to be forecast. We observe that a lower cost, more flexible solution should support *partial sensing*, where only some locations have permanent sensors, yet supporting traffic forecast at other locations currently without sensors. The partial sensing research is in its nascent stage, with limited recent work on *short-term forecast*. This paper investigates the *long-term partial sensing forecast* problem that aims to train a prediction model to use recent measurements from currently sensed locations to forecast traffic at currently unsensed locations deep into the future.

**Challenges:** The long-term partial sensing forecast problem poses several non-trivial challenges. Fig. 1 shows the traffic rates at locations 121, 44, and 33 over two days from the PEMS08 dataset. Using

---

\*Corresponding author

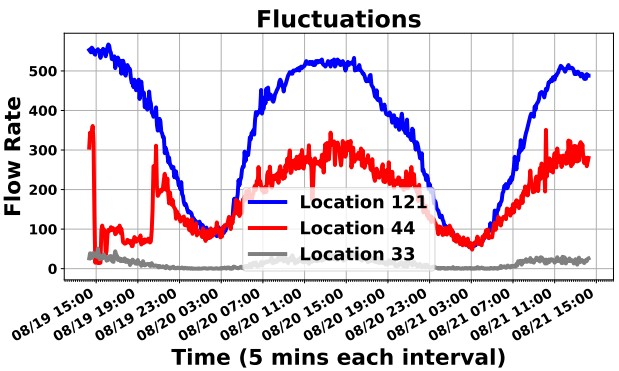

Figure 1: Flow rates of three locations for two days

past traffic measurements at one location to infer future traffic at another location requires not only the adaptability of a model to transfer the knowledge from locations to location based on the limited input features but also learning the intrinsic, subtle spatio-temperal connections across the locations. Next, long-term forecasting requires the learning of informative embeddings capable of capturing more intricate spatial and temporal correlations than what is required for short-term forecasting. We also observe high-frequency traffic fluctuations at the locations of Fig. 1, which are noises against the underlying traffic patterns. We will show that the impact of noise on traffic forecast accuracy is much larger under long-term partial sensing than under traditional short-term full sensing.

**Related Work:** Most existing work on traffic forecast focuses on *full sensing forecast*, where all locations of interest are deployed with permanent sensors, including recurrent neural networks (RNNs) (Zhang et al., 2017), convolutional neural networks (CNNs) or Multiple Layer Perceptrons (MLPs) with graph neural networks (GNNs) (Yu et al., 2017; Li et al., 2018; Wu et al., 2021b; Li & Zhu, 2021; Jiang et al., 2023b; Yi et al., 2023), neural differential equations (NDEs) (Fang et al., 2021; Liu et al., 2023c), Bayesian neural network (Miao et al., 2022), transformers (Liu et al., 2023a; Jiang et al., 2023a), mixture-of-experts (MOE) (Shazeer et al., 2017) based spatio-temporal model (Lee & Ko, 2024), and large language model (LLM) base sptiao-temporal model (Liu et al., 2024; 2025c;a;b). A few recent works study *partial sensing forecast*, where only some locations are deployed with permanent sensors, such as Frigate (Gupta et al., 2023), STSM (Su et al., 2024), and GinAR (Yu et al., 2024). But these methods are designed for short-term forecasting. The most related work is GinAR, which performs much better than Frigate, and unlike STSM, it does not require additional knowledge about location environment (such as nearby malls). But GinAR mainly focuses on missing observations and does not have a mechanism to address the impact of noise, and is therefore not suitable for long-term forecast, as we will demonstrate.

Spatial transfer learning methods (Mallick et al., 2021; Tang et al., 2022; Ouyang et al., 2023) utilize traffic measurements from data-rich locations to forecast the traffic at data-scarce locations. Data imputation methods (Li et al., 2023; Cini et al., 2021; Chen et al., 2023) interpolate the missing measurements at some locations where sensors may sometimes fail. They still require sensors at all locations. Spatial extrapolation methods, such as non-negative matrix factorization methods (Lee & Seung, 2000), and GNN methods (Wu et al., 2021a; Zheng et al., 2023; Hu et al., 2023; Appleby et al., 2020), could be used to estimate traffic at unsensed locations based on data from sensed locations within the same time period, but they do not model the complex spatio-temporal dynamics in long-term traffic forecast, and hence are not suitable for traffic forecast deep into the future.

Recent works have addressed related challenges from different perspectives, including inductive forecasting (ST-FiT (Lei et al., 2025)), OOD generalization (STONE (Wang et al., 2024)), continual learning (URCL (Miao et al., 2024), and Expand-and-Compress (Chen & Liang, 2024)), and test-time adaptation (Learning with Calibration (Chen & Liang, 2025)). ST-FiT assumes that the target sensors have no training data but all sensors are observed at inference, and focuses on short-term forecasting; our setting instead uses temporary sensors during training and assumes permanent missingness at test time, with long-term

prediction. STONE models spatial and temporal distribution shifts via dynamic node sets, while our setting assumes a fixed node set with partial sensor deployment. Expand-and-Compress introduces continual adaptation over time, whereas our framework considers a static deployment without continual evolution. Learning with Calibration performs online test-time adaptation, while our model remains frozen during inference and aims for robust generalization without test-time updates. In contrast to these works, we are the first to study long-term forecasting under permanent partial sensing, where the spatial distribution shift is both large-scale and fixed at deployment.

In summary, this is the first work specifically on the partial sensing long-term traffic forecast problem.

**Contributions:** We propose a **S**patial-temporal **L**ong-term **P**artial sensing **F**orecast model (**SLPF**) for long-term traffic forecasting. SLPF has three main technical contributions. First, we design a rank-based node embedding, which makes the model more robust against noises in learning intricate spatio-temporal correlations for long-term forecast. Second, we propose a spatial transfer module, which aims to enhance our model's adaptability by extracting the dynamic traffic patterns from the transfer weights based on rank-based embedding in addition to spatial adjacency, allowing for more nuanced and accurate predictions. Third, we use a multi-step training process to fully utilize the available training data. This approach enables successive refinement of the model parameters. Extensive experiments on several real-world traffic datasets demonstrate that our model outperforms the state-of-the-art and achieves superior accuracy in partial sensing long-term forecasting. Our source code is at `https://github.com/zbliu98/SLPF`.

## 2 Preliminaries

### 2.1 Problem Definitions

***Definition 1:*** *Road Topology and Traffic Flow Rates.* Consider a road system, and let $N$ be a set of chosen locations where traffic statistics are of interest. Let $n = |N|$. The *road topology* of these locations is represented as a graph $\mathcal{G} = (N, A)$. $A = [A_{i,j}, i, j \in N] \in \mathbb{R}^{n \times n}$ is an adjacency matrix. $A_{i,j} = 1$ indicates that there is a road between location $i$ and location $j$; otherwise $A_{i,j} = 0$. A *traffic flow* consists of all the vehicles that pass a location in $N$; the *flow rate* is defined as the number of vehicles passing through the location during a preset time interval. It is a discrete function of time if we partition the time into a series of time intervals of a certain length, e.g., 5 min. For simplicity, we normalize each time interval as one unit of time. Let $M$ be a subset of locations, i.e., $M \subseteq N$, and $T$ be a series of time intervals. As an example, $T$ could be $\{t-l+1, ..., t-1, t\}$ of $l$ intervals, where $t$ is the current time. The *traffic matrix* over locations $M$ and times $T$ is defined as $\mathcal{X}_{M,T} = [\mathcal{X}_{i,j}, i \in M, j \in T]$, where $\mathcal{X}_{i,j}$ is the flow rate at location $i$ during time $j$. Further denote the rate vector at location $i$ as $\mathcal{X}_{i,T} = [\mathcal{X}_{i,j}, j \in T]$. Hence, $\mathcal{X}_{M,T} = [\mathcal{X}_{i,T}, i \in M]$.

***Definition 3:*** *Problem of Partial-Sensing Traffic Forecast.* Suppose a subset $M$ of locations are equipped with permanent sensors to continuously measure their flow rates. The subset of locations without permanent sensors is denoted as $M' = N - M$. Let $m = |M|$ and $m' = |M'|$. The problem is to forecast a future traffic matrix $\mathcal{X}_{M',T'}$ over the locations currently without sensors, based on a recent traffic matrix $\mathcal{X}_{M,T}$ that has been just measured at the locations with sensors, where $T' = \{t+1, ..., t+l'\}$, $T = \{t-l+1, ..., t-1, t\}$, and $t$ is the current time. It is called *partial-sensing traffic forecast* if $M \subset N$ and $M' \neq \emptyset$; it is *full-sensing traffic forecast* if $M = N$. In most existing work (Fang et al., 2021; Song et al., 2020; Jin et al., 2022; Shao et al., 2022a; Jia & Benson, 2020; Ji et al., 2022), both $l$ and $l'$ are set to 1 hour for *short-term forecast*. Note that $l$ should not be too large to avoid excessively large models and computation costs that come with them. Moreover, research has shown that too large $l$ may actually degrade forecast accuracy (Zeng et al., 2023). For *long-term forecast*, which is the focus of this work, $l'$ is set much larger than $l$.

### 2.2 Embeddings

Embeddings are a common technique to enhance feature expression. We adopt the embedding setting in (Shao et al., 2022a; Liu et al., 2023a), where the time-of-day embedding and the day-of-week embedding capture the temporal information, and node embedding captures the spatial property of locations. We use $E_M^{tod} \in \mathbb{R}^{m \times d}$, $E_M^{dow} \in \mathbb{R}^{m \times d}$, and $E_M^v \in \mathbb{R}^{m \times d}$ to denote them respectively, where $d$ is the embedding

dimension. A day is modeled as $D^{tod} = 288$ discrete units of 5 minutes each. A week has $W^{dow} = 7$ days. The time-of-day is $i/D^{tod}$ for the $i$th unit in the day, and the day-of-week is $i/W^{dow}$ for the $i$th day in the week. Trainable vectors $B_i^{tod} \in \mathbb{R}^d$, $i \in [0, D^{tod})$, and $B_i^{dow} \in \mathbb{R}^d$, $i \in [0, W^{dow})$, are the time-of-day embedding bank and the day-of-week bank. In practice, for the input flow rate data $\mathcal{X}_{M,T} \in \mathbb{R}^{m \times l}$, $T = \{t - l + 1, ..., t - 1, t\}$, we only consider the time feature of flow rate data $\mathcal{X}_{M,t-l+1} \in \mathbb{R}^m$ at $t - l + 1$, yielding $E_M^{tod} \in \mathbb{R}^{m \times d}$, $E_M^{dow} \in \mathbb{R}^{m \times d}$. For the node embedding, given any location $i \in M$, the trainable node embedding bank is denoted as $B_i^v \in \mathbb{R}^d$. For the flow rate data $\mathcal{X}_{M,T}$, we obtain $E_M^v \in \mathbb{R}^{m \times d}$ from the node embedding banks $B_M^v = \{B_i^v, i \in M\}$. Similar notations are defined over $N$ (or $M'$), with $M$ and $m$ in the above notations replaced by $N$ and $n$ (or $M'$ and $m'$).

## 3 Long-term Partial Sensing Forecast with Rank-based Node Embedding

### 3.1 Impact of noise

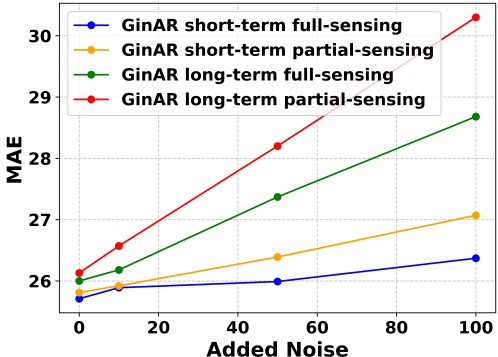

Figure 2: Noise has the greatest impact on GinAR for long-term partial sensing than for long-term full sensing, which is in turn more sensitive to noise than traditional short-term full sensing.

We find that, compared to the traditional short-term full sensing forecast, the impact of noise is exaggerated for long-term partial sensing forecast, which therefore requires special attention in the model design to handle noise impact. The "noise" in this paper refers to high-frequency fluctuations in traffic, overlaid on the underlying daily patterns as shown in Fig. 1, which may be the result from natural traffic fluctuations from one time interval to the next or sometimes from sensor misreadings. It is intuitive that noiser data, with larger fluctuations, are harder to forecast accurately. To demonstrate the impact of noise, we have performed trace-based experiments with the most related work, GinAR (Yu et al., 2024), on dataset PEMS08, where the detailed settings for all our experiments and a description of the datasets can be found in Section 5.1, and the details on more noise-related studies can be found in Section 5.4. Below we only give the most relevant information for understanding the experiment results.

Among the 170 locations in PEMS08, we randomly select half of them for the sensed locations $M$ and the other half for the unsensed locations $M'$. As the baseline for comparison, we use GinAR to perform the traditional *short-term full sensing traffic forecast* amongst the locations in $M$ only, i.e., forecasting $\mathcal{X}_{M,T'}$ from $\mathcal{X}_{M,T}$, where both $T$ and $T'$ are one hour. And we also conduct GinAR on *short-term partial sensing traffic forecast*, i.e., forecasting $\mathcal{X}_{M',T'}$ from $\mathcal{X}_{M,T}$. To quantitatively investigate the impact of noise, we add a controlled amount of noise from a zero-mean i.i.d. Gaussian distribution, $\mathcal{N}(0, \gamma^2)$, to the dataset, where $\gamma^2 \in \{0, 10, 50, 100\}$, and $\gamma^2 = 0$ represents the scenario without additional noise. The experimental results are presented as the blue line (GinAR short-term full sensing) and the orange line (GinAR short-term partial sensing) in Fig. 2, where MAE (Mean Absolute Error) is the average forecast error among all locations in $M$. It demonstrate that when the noise increase, the problem of short-term partial sensing forecasting is more challenging than short-term full sensing forecasting.

Next we move to *long-term full sensing traffic forecast*, which still forecasts $\mathcal{X}_{M,T'}$ from $\mathcal{X}_{M,T}$, but $T'$ is eight hours, while $T$ remains one hour. The experimental results are presented as the green line (GinAR long-term full sensing) in Fig. 2, which shows that noise has a larger impact on forecast accuracy in the long-term case because the error increases at a much faster rate as we increase noise. Finally we investigate the impact of noise on our new problem of *long-term partial sensing traffic forecast*, which forecasts $\mathcal{X}_{M',T'}$ from $\mathcal{X}_{M,T}$, where $M'$ is the unsensed locations, $T'$ is eight hours and $T$ is one hour. The experimental results are shown as the red line (GinAR long-term partial sensing), where the impact of noise on forecast accuracy is the largest and the error increases at the fastest rate when we increase noise. The results imply that long-term partial sensing traffic forecast is more sensitive to noise than long-term full sensing, which is in turn more sensitive to noise than traditional short-term full sensing.

The most related work, GinAR (Yu et al., 2024), does not have a component that is explicitly designed to moderate the impact of noise, which is reflected in the experimental results above. This is where our work will come in, by incorporating a new component that helps reduce the impact of noise.

## 3.2 Rank-based Node Embedding

To handle the noise impact on long-term partial sensing traffic forecast, we introduce a new *rank-based node embedding*. Consider an arbitrary time interval $j \in T$ in the input $\mathcal{X}_{M,T}$. We assign a rank value to each location $k \in M$ as follows: Sort $\mathcal{X}_{i,j}$, $\forall i \in M$, in ascending order (with ties broken arbitrarily), and the rank of location $k$ at time $j$ is the index number of $\mathcal{X}_{k,j}$'s position in the ordered list.

Consider the example of Fig. 1, location 44 (middle red curve) exhibits a daily traffic pattern with high-frequency noises as the traffic fluctuates normally or sometimes due to inaccurate sensor readings. Suppose location 44 is a sensed location, whereas location 121 (top blue curve) is unsensed. Traffic fluctuations at location 44 and location 121 blur their patterns, making it harder to use a blurred pattern at location 44 to predict the noisy traffic at location 121, especially for long-term prediction where the intrinsic connection between the two locations becomes weaker and thus the impact of noise will be bigger due to a smaller "signal-to-noise" ratio in data. Comparing to the actual, fluctuating flow rates, the ranks are discretized values that tend to smooth out noise fluctuation and be more stable. Especially for partial sensing, the fewer the number of sensed locations is, the more stable the rank-based embedding becomes. The stability of the ranks could also enhance the model's adaptability from one spatial area (say, with sensor data) to a new area (without sensor data), whereby their actual flow rates can differ greatly but their relative ranks follow stable patterns. In other words, the rank-based embedding is more robust to the distribution shift from the sensed locations to the unsensed locations.

## 3.3 Long-term Partial Sensing Forecast Model

We present our Long-term Partial sensing Forecast model (LPF) with rank-based node embedding. At each time interval, we rank the flow rates at the locations in $M$. Given the $i$-th rank of the locations, vector $B_i^r \in \mathbb{R}^d$ denotes the rank-based node embedding bank for rank $i$. For the input traffic data $\mathcal{X}_{M,T}$, we perform ranking for each time interval in $T$ over $M$ locations, yielding the rank-based node embedding $E_M^r \in \mathbb{R}^{m \times l \times d}$.

A Multi-Layer Perceptron (MLP) processes the input data $\mathcal{X}_{M,T}$ to produce a feature embedding $E_M^f \in \mathbb{R}^{m \times d}$. We then concatenate this with the other four embeddings: time-of-day embedding $E_M^{tod}$, day-of-week embedding $E_M^{dow}$, node embedding $E_M^v$, rank-based node embedding $E_M^r$, to form an aggregated feature $\in \mathbb{R}^{m \times 5d}$. This aggregated feature then passes through another MLP to produce a high-dimensional representation $H_{M,T} \in \mathbb{R}^{m \times 5d}$.

$$E_M^f = MLP(\mathcal{X}_{M,T}) \tag{1}$$

$$H_{M,T} = MLP(E_M^f || E_M^v || E_M^r || E_M^{tod} || E_M^{dow}) \tag{2}$$

Next, we want to capture the latent correlations between sensed locations and unsensed locations and to transfer the feature embedding from the sensed locations in $M$ to the unsensed locations in $M'$. For that,

we introduce the *node embedding enhanced spatial transfer matrix*, $A'_{M,M'}$, as follows:

$$A'_{M,M'} = A_{M,M'} + (E^r_M + B^v_M)(B^v_{M'}{}^\tau) \tag{3}$$

where $\tau$ is the transpose of the matrix, and $A_{M,M'} = [A_{i,j}, i \in M, j \in M'] \in \mathbb{R}^{m \times m'}$ is the partial adjacency matrix between locations in $M$ and locations in $M'$. We enhance our model's spatial correlation by incorporating the node embedding banks $B^v_M$ and the rank-based node embedding $E^r_M$, thereby producing more robust embeddings. Besides, the node embedding banks $B^v_M$ makes the enhanced spatial transfer matrix aware of the location properties. The rank-based node embedding $E^r_M$ makes the matrix sensitive to the changes in traffic patterns, and thus enhances the model's adaptability. With this enhanced spatial transfer matrix, an MLP is then used to map the high-dimensional representation, $A'_{M,M'}{}^\tau H_{M,T}$, to the traffic forecast output $\hat{\mathcal{X}}_{M',T'}$.

$$\hat{\mathcal{X}}_{M',T'} = MLP(A'_{M,M'}{}^\tau H_{M,T}) \tag{4}$$

## 4 Additional Training Data

### 4.1 Inference v.s. Training

After a forecast model is trained and deployed, the permanent sensors at the locations in $M$ will measure flow rates $\mathcal{X}_{M,T}$, which will be used as input to the model to inference (forecast) future rates of the locations in $M'$, i.e., $\mathcal{X}_{M',T'}$.

Before model deployment, when we learn such a model, we will need the ground truth of $\mathcal{X}_{M',T'}$ in the training data, which is unavoidable also for the related work including GinAR (Yu et al., 2024). One way is to use mobile sensors that are temporarily deployed at the locations in $M'$ for a period of time to collect the training data. This is reasonable because mobile sensors can be re-deployed at future new locations or in different cities for collecting the needed training data, i.e., the ground truth for the model output, offering a lower overall cost than implementing permanent sensors at all locations of all cities that need traffic forecast now or in the future.

With mobile sensors temporarily deployed to collect the traffic data at $M'$, say, for three months, we naturally have both $\mathcal{X}_{M',T'}$ and $\mathcal{X}_{M',T}$ for that period of time. So, the training dataset will naturally contain $\mathcal{X}_{M,T}$ and $\mathcal{X}_{M,T'}$ from the permanent sensors, and $\mathcal{X}_{M',T}$ and $\mathcal{X}_{M',T'}$ from the mobile sensors. During the model training phase, even though the input to the model must still be $\mathcal{X}_{M,T}$ and the output be $\mathcal{X}_{M',T'}$, we should fully utilize the side information of $\mathcal{X}_{M',T}$ and $\mathcal{X}_{M,T'}$, which is available in the training data, to refine the model parameters.

### 4.2 Spatio-Temporal Long-Term Partial Sensing Forecast Model

We propose the Spatio-temporal Long-term Partial sensing Forecast model (SLPF) to utilize the additional training data, as illustrated in Fig. 3, where the training phase (left plot) consists of three sequentially steps: 1. the *dynamic adaptive step*, which builds a module with $\mathcal{X}_{M,T}$ as input and $\mathcal{X}_{M',T}$ as expected output; 2. the *long-term forecasting step*, which builds another module with $\mathcal{X}_{M,T}$ and the previous module's output as input, and with $\mathcal{X}_{M,T'}$ as expected output; and 3. the *aggregation step*, which builds yet another module with $\mathcal{X}_{M,T}$ and the previous two modules' output as input, and with $\mathcal{X}_{M',T'}$ as expected output. The above three modules are trained sequentially, and we refer to them as the *dynamic adaptive module*, the *long-term forecasting module*, and the *aggregation module*, respectively, which together form the proposed SLPF.

### 4.3 Dynamic Adaption Step

The main objective of this module is to capture the latent correlations between the locations in $M$ and the locations in $M'$ and to transfer the feature embedding from $M$ to $M'$. We first use $\mathcal{X}_{M,T}$ through Eq. 1-2 to obtain the representation $H_{M,T}$. Then, the node embedding enhanced spatial transfer matrix $A'_{M,M'}$, as stated in Eq. 3, is used to transfer the knowledge from $M$ to $M'$, where the node embedding banks $B^v_M$,

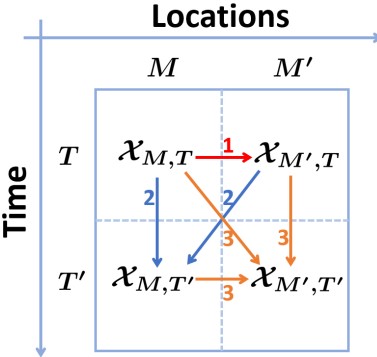

**1: Dynamic Adaption Step from** $\mathcal{X}_{M,T}$ **to** $\mathcal{X}_{M',T}$
**2: Long-Term Forecasting Step from** $\mathcal{X}_{M,T}$ , $\mathcal{X}_{M',T}$ **to** $\mathcal{X}_{M,T'}$
**3: Aggregation Step from** $\mathcal{X}_{M,T}$ , $\mathcal{X}_{M',T}$ , $\mathcal{X}_{M,T'}$ **to** $\mathcal{X}_{M',T'}$

Figure 3: The model training phase consists of three steps above. After the model is trained and deployed, only $\mathcal{X}_{M,T}$ is collected for input, and $\mathcal{X}_{M',T'}$ is output.

$B_{M'}^v$, and the rank-based node embedding $E_M^r$ are trained during this step. Finally, an MLP is used to map the high-dimensional representation, $A'_{M,M'}{}^\tau H_{M,T}$, to an output $\hat{\mathcal{X}}_{M',T}$, which is the module output for the ground truth $\mathcal{X}_{M',T}$ by the dynamic adaption step.

### 4.4 Long-Term Forecasting Step

The main objective of this module is to enhance SLPF's long-term forecasting power, by learning informative embeddings capable of capturing more intricate spatial and temporal correlations than what's needed for short-term forecasting. The input consists of $\mathcal{X}_{M,T}$ and $\hat{\mathcal{X}}_{M',T}$ which is the output of the previous step. We first produce a rank embedding enhanced spatial transfer matrix, similar to Eq. 3, as follows:

$$A'_{N,M} = A_{N,M} + (E_N^r + B_N^v)(B_M^v{}^\tau) \tag{5}$$

where $A_{N,M} = [A_{i,j}, i \in N, j \in M] \in \mathbb{R}^{n \times m}$ is the partial adjacency matrix between locations in $N$ and locations in $M$, the rank embedding $E_N^r \in \mathbb{R}^{n \times d}$ is obtained from $\mathcal{X}'_{N,T} = [\mathcal{X}_{M,T}, \hat{\mathcal{X}}_{M',T}]$, using the method described in Section 3.2, and $B_N^v$ and $B_M^v$ are defined in Section 2.2. The representation $H_{N,T} \in \mathbb{R}^{n \times 5d}$ is obtained from $\mathcal{X}'_{N,T} \in \mathbb{R}^{n \times l}$, similar to Eq. 1-2, as follows:

$$E_N^f = MLP(\mathcal{X}'_{N,T}) \tag{6}$$

$$H_{N,T} = MLP(E_N^f||E_N^v||E_N^r||E_N^{tod}||E_N^{dow}) \tag{7}$$

Finally, an MLP is used to produce $\hat{\mathcal{X}}_{M,T'}$ of length $l'$, as the module output for the ground truth $\mathcal{X}_{M,T'}$.

$$\hat{\mathcal{X}}_{M,T'} = MLP'(A'_{N,M}{}^\tau H_{N,T}) \tag{8}$$

### 4.5 Aggregation Step

So far, the model has learned how to transfer knowledge from $M$ to $M'$ during the dynamic adaption step and how to perform long-term forecasting in the long-term forecasting step. This aggregation step aggregates them together for long-term forecast of $\mathcal{X}_{M',T'}$. Its input comprises three parts, $\mathcal{X}_{M,T}$, $\hat{\mathcal{X}}_{M',T}$ from the dynamic adaption step, and $\hat{\mathcal{X}}_{M,T'}$ from the long-term forecasting step. Let $\hat{\mathcal{X}}_{N,T}$ be the concatenation of $\mathcal{X}_{M,T}$ and $\hat{\mathcal{X}}_{M',T}$. We first produce two rank embedding enhanced spatial transfer matrices, similar to Eq. 3, as follows:

$$A'_{N,M'} = A_{N,M'} + (E_N^r + B_N^v)(B_{M'}^v{}^\tau) \tag{9}$$

$$A'_{M,M'} = A_{M,M'} + (E_M^r + B_M^v)(B_{M'}^v{}^\tau) \tag{10}$$

Table 1: Basic statistics of the datasets used in our experiments.

| Dataset | PEMS03 | PEMS04 | PEMS08 | PEMS-BAY | METR-LA |
|---|---|---|---|---|---|
| Area | in CA, USA | | | | |
| Time Span | 9/1/2018 - 11/30/2018 | 1/1/2018 - 2/28/2018 | 7/1/2016 - 8/31/2016 | 1/1/2017 - 5/31/2017 | 3/1/2012 - 6/30/2012 |
| Time Interval | 5 min | | | | |
| Number of Locations, $n$ | 358 | 307 | 170 | 325 | 207 |
| Number of Time Intervals | 26,208 | 16,992 | 17,856 | 52,116 | 34,272 |

where $A_{N,M'}$ and $A_{M,M}$ are adjacency matrices, while $E_N^r$ and $E_M^r$ are the rank-based embeddings. The representations, $H_{N,T} \in \mathbb{R}^{n \times 5d}$ and $H_{M,T'} \in \mathbb{R}^{m \times 5d}$, are obtained from $\hat{\mathcal{X}}_{N,T}$ and $\hat{\mathcal{X}}_{M,T'}$, respectively, similar to Eq. 1-2. Finally, an MLP is used to produce $\hat{\mathcal{X}}_{M',T'}$ of length $l'$, as the output for the ground truth $\mathcal{X}_{M',T'}$.

$$\hat{\mathcal{X}}_{M',T'} = \alpha * MLP'(A_{N,M'}'^{\top} H_{N,T}) + (1-\alpha) * MLP'(A_{M,M'}'^{\top} H_{M,T'})$$

### 4.6 Complexity Analysis

We provide the parameter complexity and the time below.

**Parameter complexity:** Let $n$ be the number of locations (and ranks), $d$ the embedding dimension size, $D = 5d$ the aggregated dimension size, and $\alpha$ the number of MLP layers. The parameter complexity is:

$$\mathcal{O}\left((2n + N^{dow} + N^{tod}) \cdot d + n^2 + \alpha \cdot D^2\right)$$

**Time complexity:** SLPF has three stages: adaptation, forecast, and aggregation. The dominant terms are:

$$\mathcal{O}_{\text{adp}} = \underbrace{m \cdot l \cdot D}_{\text{input embedding}} + \underbrace{m \cdot D^2}_{\text{MLP encoder}} + \underbrace{m \cdot m' \cdot D}_{\text{transfer matrix}} + \underbrace{m' \cdot D \cdot l}_{\text{final MLP}}$$

$$\mathcal{O}_{\text{fore}} = \underbrace{n \cdot l \cdot D}_{\text{input embedding}} + \underbrace{n \cdot D^2}_{\text{MLP encoder}} + \underbrace{n \cdot m \cdot D}_{\text{transfer matrix}} + \underbrace{m' \cdot D \cdot l'}_{\text{final MLP}}$$

$$\mathcal{O}_{\text{agg}} = \underbrace{(n \cdot l + m \cdot l') \cdot D}_{\text{embedding from previous predictions}} + \underbrace{(n+m) \cdot D^2}_{\text{MLP encoder}} + \underbrace{(n+m) \cdot m' \cdot D}_{\text{transfer matrix}} + \underbrace{m' \cdot D \cdot l'}_{\text{final MLP}}$$

## 5 Experiment and Evaluation

### 5.1 Settings and Datasets

#### 5.1.1 Datasets

We use five widely used public traffic flow datasets, METRA-LA, PEMSBAY, PEMS03, PEMS04 and PEMS08 [1]. The details of data statistics are shown in Table 1. All datasets measure the flow rates, i.e., number of passing vehicles at each location during each 5-min interval. We split each dataset into three subsets in 3:1:1 ratio for training, validation, and testing. A traffic forecast model uses 1 hour of flow-rate data to predict the future 8 hours of flow-rate data. Because the time interval for flow-rate measurement is five minutes, each hour has 12 intervals. We pre-process the flow rates by the z-score normalization before using them as model input. The z-score normalization subtracts the mean rate from each flow rate in a dataset and then divides the result by the rates' standard deviation.

---

[1]The datasets are provided in the STSGCN github repository at `https://github.com/Davidham3/STSGCN/`, and DCRNN github repository at `https://github.com/liyaguang/DCRNN`.

Table 2: Performance comparison over five datasets, where $n$ is the total number of locations and $m'$ is the number of unsensed locations. A baseline model with suffix * is the model adapted for the partial sensing task. Our models are in bold at the bottom.

| Models | PEMS03 $m'=250, m'/n=69\%$ | | | PEMS04 $m'=250, m'/n=81\%$ | | | PEMS08 $m'=150, m'/n=88.2\%$ | | | PEMSBAY $m'=250, m'/n=76.9\%$ | | | METRLA $m'=150, m'/n=72.4\%$ | | |
|---|---|---|---|---|---|---|---|---|---|---|---|---|---|---|---|
| | MAE | RMSE | MAPE | MAE | RMSE | MAPE | MAE | RMSE | MAPE | MAE | RMSE | MAPE | MAE | RMSE | MAPE |
| Matrix Factorization (Lee & Seung, 2000) | 69.01 | 110.17 | 135.41 | 91.38 | 150.09 | 129.65 | 76.01 | 132.64 | 85.13 | 6.25 | 15.63 | 20.21 | 16.72 | 34.21 | 49.83 |
| PatchTST* (Nie et al., 2022) | 57.71 | 86.33 | 103.23 | 58.28 | 86.15 | 54.99 | 43.90 | 64.68 | 28.59 | 4.58 | 8.66 | 12.50 | 11.67 | 21.81 | 26.86 |
| iTransformer* (Liu et al., 2023b) | 67.31 | 103.07 | 127.88 | 87.44 | 125.04 | 99.1 | 69.81 | 91.03 | 61.24 | 5.13 | 9.60 | 15.62 | 13.65 | 22.58 | 27.51 |
| D2STGNN* (Shao et al., 2022c) | 37.31 | 61.1 | 45.69 | 47.29 | 75.41 | 48.98 | 47.37 | 67.4 | 35.2 | 6.62 | 10.94 | 15.52 | 14.93 | 27.25 | 42.33 |
| FourierGNN (Yi et al., 2023) | 35.55 | 52.90 | 60.43 | 43.46 | 62.47 | 43.99 | 44.97 | 62.66 | 34.13 | 3.83 | 7.36 | 10.87 | 7.80 | 14.34 | 36.41 |
| STSGCN* (Song et al., 2020) | 28.57 | 45.90 | 39.17 | 32.70 | 49.02 | 27.03 | 31.74 | 49.00 | 21.47 | 3.12 | 6.48 | 7.87 | 4.77 | 9.37 | 15.26 |
| STGODE* (Fang et al., 2021) | 23.78 | 38.69 | 32.40 | 27.81 | 42.45 | 25.53 | 33.01 | 51.20 | 24.20 | 3.15 | 6.22 | 7.76 | 6.48 | 11.49 | 19.79 |
| ST-SSL* (Ji et al., 2023) | 24.98 | 42.61 | 36.15 | 27.71 | 44.38 | 22.07 | 30.92 | 47.65 | 22.39 | 3.15 | 6.31 | 7.86 | 5.96 | 12.98 | 14.82 |
| DyHSL* (Zhao et al., 2023) | 25.12 | 45.22 | 30.19 | 29.17 | 46.44 | 22.03 | 29.10 | 45.59 | 21.87 | 3.68 | 7.67 | 10.06 | 8.14 | 16.15 | 25.85 |
| STID* (Shao et al., 2022a) | 21.70 | 39.67 | 24.26 | 24.17 | 40.76 | 17.14 | 24.99 | 42.77 | 15.42 | 3.00 | 5.98 | 7.43 | 6.96 | 14.00 | 19.68 |
| MegaCRN* (Jiang et al., 2023c) | 21.93 | 40.32 | 28.30 | 24.15 | 39.2 | 18.15 | 29.41 | 47.57 | 20.58 | 3.09 | 6.26 | 7.42 | 5.50 | 10.36 | 18.42 |
| STEP* (Shao et al., 2022b) | 21.84 | 39.91 | 25.82 | 23.50 | 39.14 | 17.40 | 28.35 | 46.51 | 17.30 | 4.84 | 9.13 | 13.01 | 7.1 | 12.51 | 25.75 |
| TESTAM* (Lee & Ko, 2024) | 23.54 | 43.12 | 27.62 | 24.11 | 39.13 | 17.11 | 26.47 | 44.43 | 16.43 | 3.15 | 6.31 | 7.40 | 4.28 | 8.66 | 14.05 |
| STLLM* (Liu et al., 2024) | 27.62 | 44.99 | 38.69 | 31.95 | 48.12 | 26.85 | 30.15 | 48.13 | 20.94 | 3.03 | 6.30 | 7.51 | 4.50 | 9.26 | 15.04 |
| STLLM+* (Liu et al., 2025a) | 27.50 | 44.98 | 39.01 | 31.92 | 48.20 | 26.93 | 30.09 | 48.19 | 20.76 | 3.02 | 6.28 | 7.47 | 4.53 | 9.30 | 15.10 |
| STAEFormer* (Liu et al., 2023a) | 22.47 | 41.60 | 26.09 | 23.11 | 38.32 | 16.35 | 23.31 | 41.25 | 14.63 | 2.72 | 5.58 | 6.50 | 4.07 | 8.43 | 13.05 |
| STAEFormer + IGNNK (Wu et al., 2021a) | 23.01 | 42.32 | 27.12 | 23.97 | 39.21 | 17.09 | 23.99 | 43.15 | 15.32 | 2.80 | 5.90 | 6.89 | 4.32 | 8.78 | 14.21 |
| STAEFormer + STGNP (Hu et al., 2023) | 22.59 | 43.15 | 26.61 | 23.10 | 38.40 | 16.39 | 23.37 | 41.39 | 14.91 | 2.73 | 5.60 | 6.55 | 4.19 | 8.58 | 13.29 |
| BiTGraph (Chen et al., 2023) | 24.14 | 44.80 | 35.26 | 24.75 | 42.01 | 17.91 | 27.62 | 46.83 | 17.67 | 3.29 | 6.50 | 7.83 | 4.43 | 8.96 | 15.75 |
| GinAR (Yu et al., 2024) | 24.11 | 44.03 | 29.28 | 24.98 | 41.07 | 19.90 | 26.91 | 47.13 | 18.21 | 3.01 | 6.91 | 7.50 | 4.29 | 9.09 | 15.13 |
| **LPF** | **21.24** | **37.10** | **25.01** | **22.01** | **38.99** | **17.01** | **23.91** | **41.19** | **15.01** | **2.64** | **5.54** | **6.35** | **3.71** | **7.26** | **11.03** |
| **SLPF** | **18.41** | **33.51** | **20.69** | **21.19** | **37.07** | **15.49** | **22.18** | **37.96** | **14.14** | **2.31** | **4.71** | **5.59** | **3.31** | **6.61** | **9.45** |

### 5.1.2 Hyperparameters

The numbers of sensed locations $m$ and unsensed locations $m'$ vary in our experiments, with $m + m' = n$. The number $l$ of time intervals in model input is 12, and the number $l'$ of time intervals in model output is 96. For the embedding parameters, $N^{dow} = 7$, $N^{tod} = 288$, and the dimension is $d = 64$. We use two layers of CNN with residual connection as the MLP structure in each of the three steps in Fig. 3. The input passes through one layer of CNN, the relu activation function (Agarap, 2018), the dropout layer (Srivastava et al., 2014) with 0.15 dropout rate, and then the second layer of CNN. A residual connection (Szegedy et al., 2017) of the original input then is added to the result for the final output. $\alpha$ in the aggregation step is 0.5. During training, we set the batch size at 64, the learning rate at $10^{-3}$, and the weight decay at $10^{-3}$ for all datasets. The optimizer is AdamW (Loshchilov & Hutter, 2019). We use Mean Absolute Error (MAE) as the loss function.

### 5.1.3 Baselines

The baselines for comparison can be found in Table 2. We adapt the existing full-sensing forecast models for partial sensing forecast by adding fully connected layers to map the feature dimensions from the sensed locations to the unsensed locations. For the short-term forecast models, the output length is increased from 1 hour to 8 hours. These adapted models are identified with a suffix *. For the spatial extrapolation models, in order to use them in the context of partial sensing traffic forecast, we need to first apply a full sensing forecast model (such as STAEFormer (Liu et al., 2023a)) that uses $\mathcal{X}_{M,T}$ to forecast $\mathcal{X}_{M,T'}$ and then use the spatial extrapolation models to estimate $\mathcal{X}_{M',T'}$ from $\mathcal{X}_{M,T'}$. We have excluded from our comparison the models that either performed very poorly (such as Frigate (Gupta et al., 2023) which performed worse than all above baselines) or could not be adapted in our experimental setting (such as STSM (Su et al., 2024) which uses the environmental knowledge of each location such as nearby shopping malls). For all baselines, we executed their original codes and adhered to their recommended configurations. The reported results represent the average outcomes from four separate experiments with four random seeds, where under the same random seed.

Table 3: Different ablation methods under weighted selection and $m' = 50$ condition on PEMS08 dataset over the average horizon.

| Method | MAE | RMSE | MAPE |
|---|---|---|---|
| SLPF | 16.58 | 29.22 | 17.85 |
| LPF (1 step, see Fig. 3, from $\mathcal{X}_{M,T}$ to $\mathcal{X}_{M',T'}$) | 18.47 | 32.08 | 18.83 |
| 2 Step (from $\mathcal{X}_{M,T}$ to $\mathcal{X}_{M',T}$, then from $\mathcal{X}_{M,T}, \mathcal{X}_{M',T}$ to $\mathcal{X}_{M',T'}$) | 17.24 | 30.18 | 18.06 |
| Plain Spatial Transfer Matrix | 17.80 | 32.35 | 17.87 |
| No Spatial Transfer Matrix | 17.76 | 32.35 | 17.85 |
| No Rank-based Node Embedding | 17.39 | 31.95 | 17.89 |

### 5.1.4   Location Selection

We use two methods for the selection of sensed/unsensed locations: *random selection* and *weighted selection*. In random selection, the sensed/unsensed locations are randomly chosen. In weighted selection, the probability of each location to be sensed is proportion to its flow rate.

### 5.1.5   Accuracy Metrics

We evaluate the traffic forecast accuracy of the proposed work and the baseline models by Mean Absolute Error (MAE), Root Mean Squared Error (RMSE), and Mean Absolute Percentage Error (MAPE). Given the ground truth $\mathcal{X}_{i,j}$ and the forecast result $\hat{\mathcal{X}}_{i,j}$, $i \in M'$, $j \in T'$, MAE at time $j = \frac{1}{m'} \sum_{i=1}^{m'} \left| \mathcal{X}_{i,j} - \hat{\mathcal{X}}_{i,j} \right|$, MAPE at time $j = \left( \frac{1}{m'} \sum_{i=1}^{m'} \left| \frac{\mathcal{X}_{i,j} - \hat{\mathcal{X}}_{i,j}}{\mathcal{X}_{i,j}} \right| \right) * 100\%$,    and RMSE at time $j = \sqrt{\frac{1}{m'} \sum_{i=1}^{m'} \left( \mathcal{X}_{i,j} - \hat{\mathcal{X}}_{i,j} \right)^2}$.

### 5.2   Traffic Forecast Accuracy

Table 2 compares our methods, LPF and SLPF, with the baselines on traffic forecast accuracy in terms of average MAE, MAPE, and RMSE over 96 future intervals (i.e., 8 hours). All experiments were conducted using the weighted selection method. The number of unsensed locations is set to 250 for larger datasets including PEMS03, PEMS04 and PEMSBAY, and 150 for PEMS08 and METRLA, which have fewer locations. The percentage of locations that are unsensed ranges from 69% to 88.2%. The table shows that our model outperforms all baseline models in traffic forecast accuracy, for example, achieving 15.5% improvement by LPF and 28% improvement by SLPF in MAPE against the best baseline result over the METRLA dataset.

The long-term traffic forecast models, PatchTST* and iTransformer*, do not sufficiently model spatial correlation patterns, thus resulting in relatively poor performance. Among all the baseline models, STAEFormer* — which is a spatio-temporal short-term forecast model — achieves the best overall performance, benefiting from its sophisticated embedding techniques (e.g., node embedding, time-of-day and day-of-week embedding), coupled with a robust transformer structure. TESTAM* performs close to STAEFormer*, thanks to its informative MOE architecture. The spatial extrapolation models, IGNNK and STGNP, were originally designed for extrapolation to a relatively modest extent, such as 30%, in contrast to our experimental setting of extrapolation to 69-88.2% of locations that are unsensed from only 12.8-31% of locations that are sensed. In addition, they use STAEFormer for extrapolating in the temporal dimension from one hour data to eight hours data. In comparison, our model LPF performs better than all baselines as expected because this is the first work designed specifically for the problem of long-term partial sensing traffic forecast with a noise mitigating component of rank-based embedding. SLPF performs better than LPF with a novel three-steps model design to take advantage of additional training data.

### 5.3   Ablation Study

To investigate the effect of different components of SLPF, we conduct ablation experiments on PEMS08 with several different variants of our model. **LPF**: It does not use the additional training data in the training

phase. **2-Steps:** This variant of SLPF removes the long-term forecasting step but keeps the other two steps. It first produces an estimate of $\mathcal{X}_{M',T}$ in the dynamic adaption step and then integrates $\mathcal{X}_{M,T}$ and the estimate of $\mathcal{X}_{M',T}$ in the aggregation step to forecast $\mathcal{X}_{M',T'}$. **Plain Spatial Transfer Matrix:** We retain SLPF's 3-steps structure, but use a learnable matrix to add up with the partial adjacency matrix, instead of the node embedding bank $B^v$ and the rank-based node embedding $E^r$. **No Spatial Transfer Matrix:** This variant excludes the spatial transfer matrix of SLPF. Instead, it employs a single linear layer of MLP with shape of $\mathbb{R}^{M \times M'}$ to map the input sensor dimensions to the desired output dimensions, akin to the method we used for adapting the full sensing models for the partial sensing task. **No Rank-based Node Embedding:** We remove the rank-based node embedding, while keeping all other embeddings as well as the 3-steps structure.

Table 3 compares SLPF, LPF, and other variants above in terms of average MAE, MAPE and RMSE over 8-hour traffic forecast on dataset PEMS08, under the weighted selection of unsensed locations with $m' = 50$. Comparing SLPF (3 steps) with LPF (1 step) and 2-Steps, we can see that our 3-steps training process keeps improving forecast accuracy as more supervised information is used. Comparing SLPF with No Rank-based Node Embedding, we show that the rank-based node embedding improves the average forecast accuracy. The significance of the ranked-based embedding will be further revealed in the next subsection when we take a deeper look at its impact. Comparing SLPF with Plain Spatial Transfer Matrix and No Spatial Transfer Matrix, we show that our node embedding enhanced spatial transfer matrix is more effective than simple or no spatial transfer matrix. From Table 3, as each variant disables one design component of SLPF, we observe about 1-2% MAE degradation, which is non-negligible in this line of research, as seen from Table 3. Hence, from all above variants, the ablation study justifies the importance of all those design components, which contribute to SLPF in different ways: rank improves robustness against noise, transfer matrix enhances spatial generalization, and temporal embeddings capture periodic patterns, etc.

### 5.4   Rank-based Node Embedding

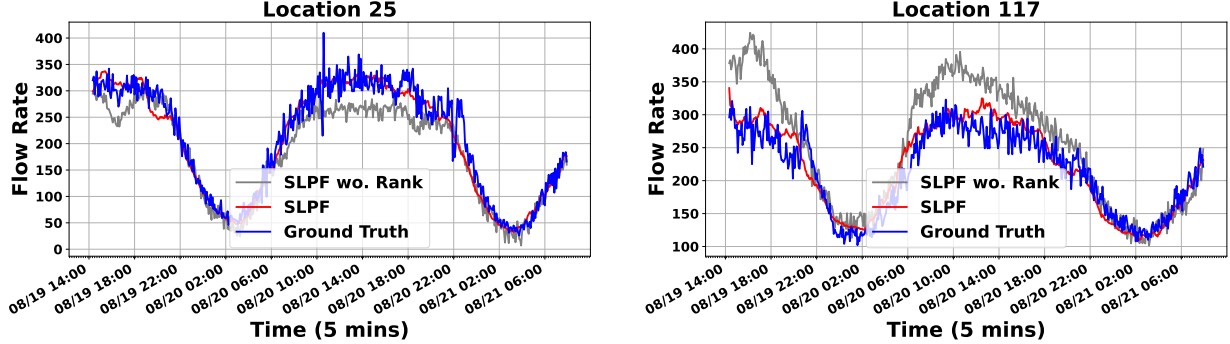

Figure 4: Visualization on different locations when our model with or without ranking embedding.

We begin with two case studies to visualize the impact of rank-based node embedding. Fig. 4 shows the actual flow rates (blue), the forecast rates of SLPF (red), and the forecast rate of SLPF without rank embedding (gray) at Location 25 and Location 117 in the PEMS08 dataset. It clearly demonstrates that rank embedding allows the red line of SLPF to track the underlying pattern by smoothing out the noise (traffic fluctuations) of the actual rate curve, outperforming the gray line of SLPF without rank embedding. We hypothesized in Section 3.2 that node ranks, which are essentially the normalized, discrete flow rates at the nodes, are more stable input to the model than the actual rates, which fluctuate. Hence, rank embedding is more resistant against noise in traffic, and thus helps capture the underlying patterns better. Below, we use experiments (Fig. 5, 7, 6) to verify our hypothesis.

To quantitatively study the robustness of rank-based node embedding against noise, we add controlled amounts of zero-mean i.i.d. noise to the training dataset, considering three commonly used distributions:

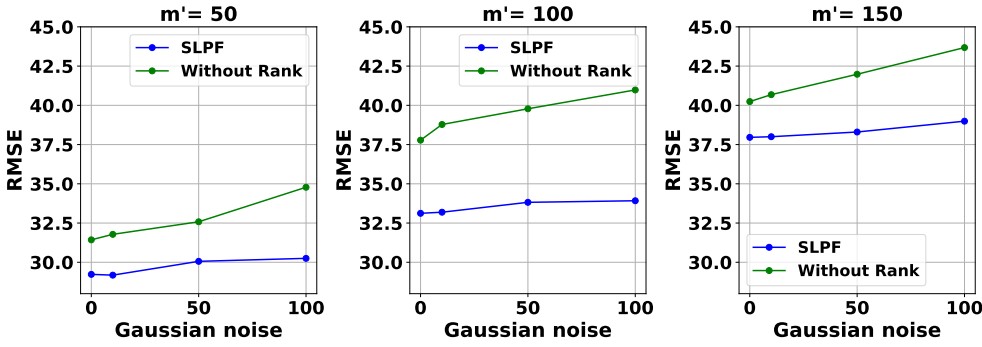

Figure 5: Performance comparison with varying $\gamma$ under Gaussian noise on PEMS08 dataset.

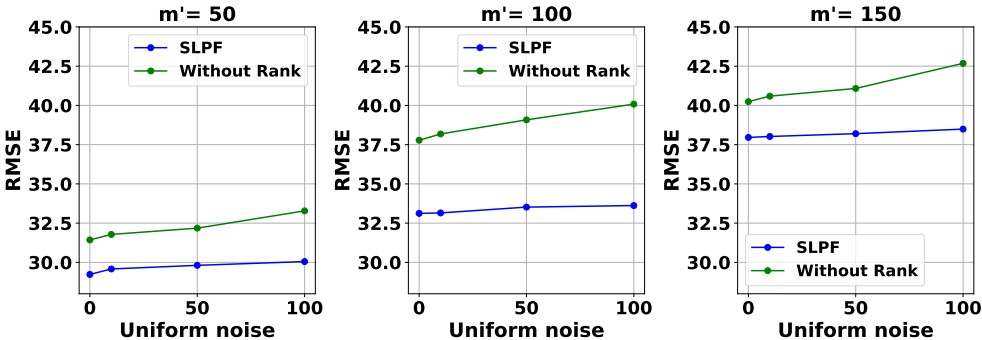

Figure 6: Performance comparison with varying $\gamma$ under Uniform noise on PEMS08 dataset.

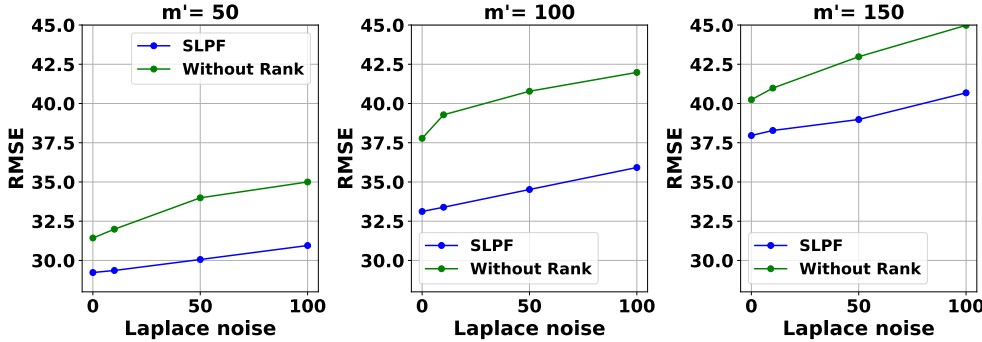

Figure 7: Performance comparison with varying $\gamma$ under Laplace noise on PEMS08 dataset.

- **Gaussian noise** $\mathcal{N}(0, \gamma^2)$:

$$p(x) = \frac{1}{\sqrt{2\pi\gamma^2}} \exp\left(-\frac{x^2}{2\gamma^2}\right)$$

- **Uniform noise** $\mathcal{U}(-\gamma, \gamma)$:

$$p(x) = \begin{cases} \frac{1}{2\gamma}, & \text{if } x \in [-\gamma, \gamma] \\ 0, & \text{otherwise} \end{cases}$$

- **Laplace noise** $\text{Laplace}(0, \gamma)$:

$$p(x) = \frac{1}{2\gamma} \exp\left(-\frac{|x|}{\gamma}\right)$$

For all three distributions, the noise magnitude is controlled by a shared scale parameter $\gamma$, where $\gamma \in \{0, 10, 50, 100\}$. The case $\gamma = 0$ corresponds to the scenario without additional noise.

The experimental results under these noise types are presented in Fig. 5, Fig. 6, and Fig. 7. Across all cases, SLPF without rank-based node embedding suffers significant performance degradation as the noise level increases. In contrast, SLPF with rank-based node embedding consistently demonstrates strong robustness, exhibiting much smaller performance drops under increasing noise levels, regardless of the noise distribution.

## 5.5 Efficiency Study

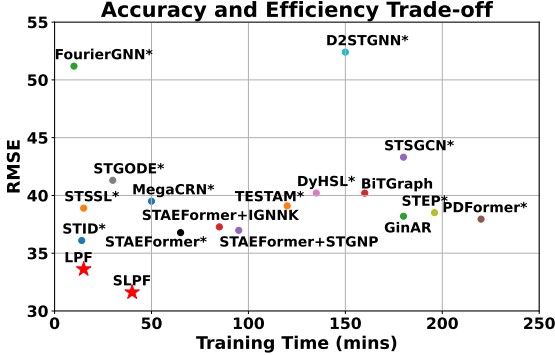

Figure 8: Comparing our model SLPF with the baselines in training efficient, in the context of accuracy-efficiency tradeoff, on dataset PEMS08, with weighted selection and $m' = 50$.

Fig. 8 compares our models, LPF and SLPF, with other models in the context of efficiency-accuracy tradeoff. Experiments were conducted on a server with AMD EPYC 7742 64-Core Processor @ 2.25 GHz, 500 GB of RAM, and NVIDIA A100 GPU with 80 GB memory. The batch size is uniformly set to 16. We record the total training times and the average RMSE values over 8 hours of traffic forecast on dataset PEMS08 with $m' = 50$. Our models achieve the best RMSE among all models with relatively low computational cost. This dual advantage of high accuracy and efficiency sets our models apart in the realm of partial sensing traffic forecast. On the one hand, STID*, FourierGNN*, STGODE*, and STSSL* exhibit commendable efficiency, yet they fall short in achieving the accuracy level demonstrated by our models. On the other hand, MegaCRN*, TESTAM*, STAEFormer*, STAEFormer+IGNNK, STAEFormer+STGNP, GinAR, BiTGraph, STSGCN*, DyHSL*, D2STGNN*, STEP*, and PDFormer* show slower training speed and worse accuracy than our models. The balance of efficiency and accuracy makes our models attractive for real-world applications in intelligent traffic management. Between LPF and SLPF, the former is less accurate but more efficient, while the latter is more accurate but less efficient, representing an accuracy-efficiency tradeoff.

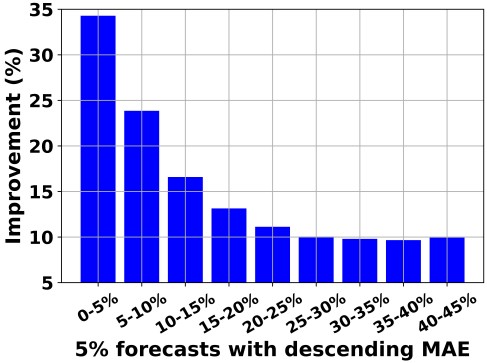

Figure 9: Improvement of SLPF over its variant without rank in bins of 5% forecasts with descending MAE, on dataset PEMS08 dataset with weighted selection and $m' = 50$.

## 5.6 Worst-case Improvement

With rank-based embedding to handle the impact of noise, particularly, large noise, we expect to observe more worst-case accuracy improvement than the average-case improvement in Table 3. To verify this hypothesis, we sort all 8 hours of traffic forecasts in the descending order of MAE, and then group them into 20 bins, each with 5% of all forecasts. We compute the average MAE in each bin for SLPF, do the same for its variant without rank, and then compute the percentage improvement in average MAE of SLPF over its variant without rank. The results are presented in Fig. 9, where SLPF consistently improves over its variant without rank over all bins shown. The improvement amongst the first 5% percent forecasts (i.e., worse case) is close to 35%, where the largest irregularities are suspected to reside.

## 5.7 Number of Unsensed Locations and Random Selection vs Weighted Selection

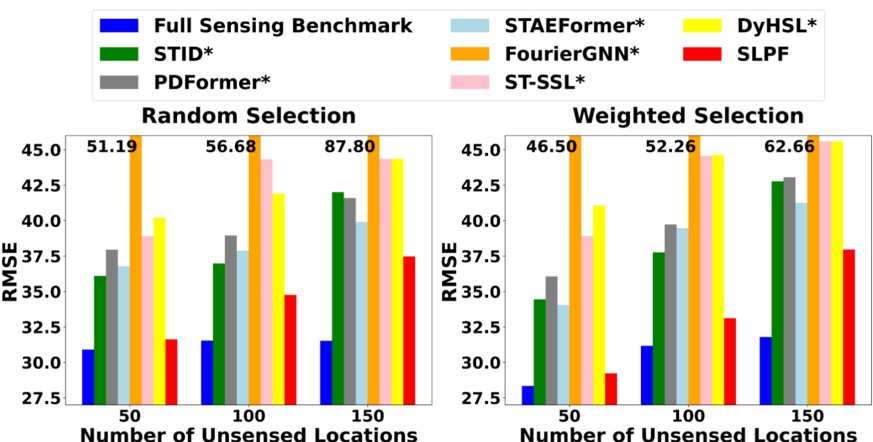

Figure 10: Accuracy comparison in terms of RMSE with respect to the number of unsensed locations under different selection methods on PEMS08 dataset.

Fig. 10 compares our model SLPF with the baselines on forecast accuracy in terms of RMSE over dataset PEMS08 with respect to a varying number of unsensed locations from 50 to 100 to 150. The left plot uses the random selection method, and the right plot uses the weighted selection method. In each plot, for each number of unsensed locations on the horizontal axis, a group of bars present the average RMSE values of the Full Sensing Benchmark (leftmost), STID*, ..., and our method SLPF (rightmost), respectively.

The Full Sensing Benchmark tries to establish some sort of accuracy bound that a partial sensing model such as SLPF can possibly achieve. We reason that a partial sensing model without the recent knowledge of unsensed locations, i.e., $\mathcal{X}_{M',T}$, should not beat a full sensing model with the knowledge of $\mathcal{X}_{M',T}$. This knowledge of unsensed locations is hypothetical, and thus the Full Sensing Benchmark is not implementable, but it gives us an accuracy bound for a partial sensing model. Table 4 shows the forecast accuracy of the full-sensing baselines on dataset PEMS08, assuming the knowledge of $\mathcal{X}_{M',T}$, such that they can use $\mathcal{X}_{N,T}$ to infer $\mathcal{X}_{M',T'}$, where $N = M + M'$. Among the full-sensing baselines, PDFormer performs the best in RMSE and thus its results are used as the Full Sensing Benchmark in Fig. 10.

Table 4: Performance comparison on PEMS08 dataset among the full-sensing baselines with the hypothetical assumption that they have the knowledge of the unsensed locations. The best results are in bold.

| Models | PEMS08 | | |
|---|---|---|---|
| | MAE | RMSE | MAPE |
| STID (Shao et al., 2022a) | 19.59 | 32.20 | 13.05 |
| PDFormer (Jiang et al., 2023a) | **18.58** | **31.86** | 12.64 |
| STAEFormer (Liu et al., 2023a) | 18.77 | 32.73 | **12.10** |
| TESTAM (Lee & Ko, 2024) | 18.90 | 32.81 | 12.45 |
| MegaCRN (Jiang et al., 2023c) | 19.99 | 33.67 | 13.09 |
| STEP (Shao et al., 2022b) | 18.97 | 32.98 | 12.66 |
| D2STGNN (Shao et al., 2022c) | 34.50 | 46.13 | 29.96 |
| FourierGNN (Yi et al., 2023) | 43.71 | 62.66 | 38.70 |
| ST-SSL(Ji et al., 2023) | 24.35 | 37.31 | 17.59 |
| DyHSL(Zhao et al., 2023) | 26.04 | 39.28 | 20.09 |

Note that (1) the vertical axis of Fig. 10, RMSE, begins from 27.5, not zero, and (2) it ends at 45.0, while the higher bars (which go beyond the figure) have their heights shown at the top of the figure.

First, the figure shows that the choice between the random selection method and the weighted selection method does not cause significant difference in forecast accuracy. It suggests that preference of installing sensors at locations of larger flow rates does not improve performance significantly over random selection of sensor locations.

Second, our model SLPF outperforms the baselines under different numbers of unsensed locations, consistent with the results in Table 2. Although the performance of SLPF is worse than the Full Sensing Benchmark (as is expected), when the ratio of unsensed locations to all locations is relatively low, e.g., $m' = 50$ and $m'/n = 50/170 = 29.4\%$, the accuracy of SLPF is very close to the Full Sensing Benchmark. Even when the ratio of unsensed locations to all locations is high, e.g., $m' = 150$ and $m'/n = 150/170 = 88.2\%$, SLPF is about 10% less accurate than the Full Sensing Benchmark. This comparison demonstrates the validity of the partial sensing approach in traffic forecasting: By installing a significantly smaller number of permanent sensors (e.g., $m = n - m' = 170 - 150 = 20$), we can achieve long-term traffic forecasting accuracy within about 10% of what the full sensing models can achieve under a much more expensive deployment of installing sensors at all locations.

## 5.8   Forecast Length

Our partial sensing long-term traffic forecast experiments output traffic estimates for 8 hours (i.e., 96 time intervals) into the future. The *forecast length* refers to a specific time interval in the output. For example, when we consider a forecast length of 1 hour (i.e., the 12th interval), we refer to the traffic estimates for the last interval of the first hour into the future.

Fig. 11 compares our method with the baselines and the hypothetical Full Sensing Benchmark on forecast accuracy in terms of RMSE over dataset PEMS08 with respect to a varying forecast length of 1, 2, 4, and 8 hours. The figure shows that our method significantly outperforms all baselines at different forecast lengths. This result complements the average accuracy comparison in Table 2 by providing more detailed comparison in forecast accuracy over time. Interestingly, the accuracy of our method converges towards its accuracy

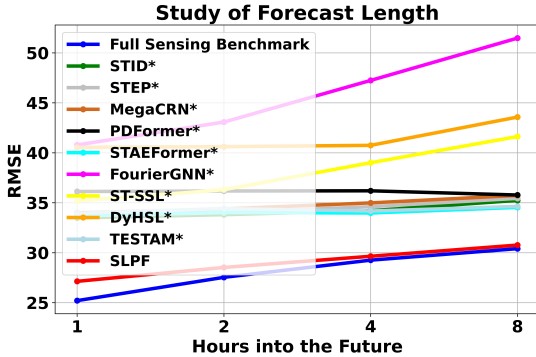

Figure 11: Accuracy comparison in terms of RMSE with respect to different forecasting lengths on PEMS08 dataset with the random selection method and the number of unsensed location being $m' = 50$.

bound of the hypothetical Full Sensing Benchmark as we stretch the forecast length from 1 hour to 8 hours, suggesting the suitability of SLPF as a long-term traffic estimator.

## 5.9 Effectiveness of Rank-based Node Embedding

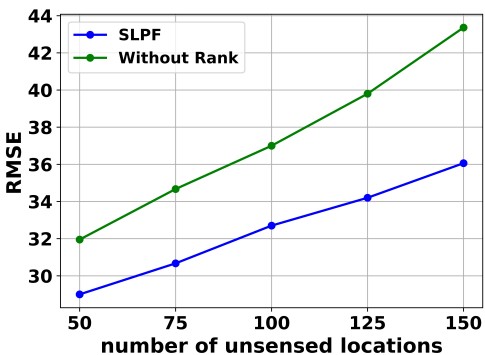

Figure 12: Impact of number of unsensed locations on forecast accuracy (RMSE) of SLPF when it is with or without rank-based node embedding, on PEMS08 dataset with weighted selection.

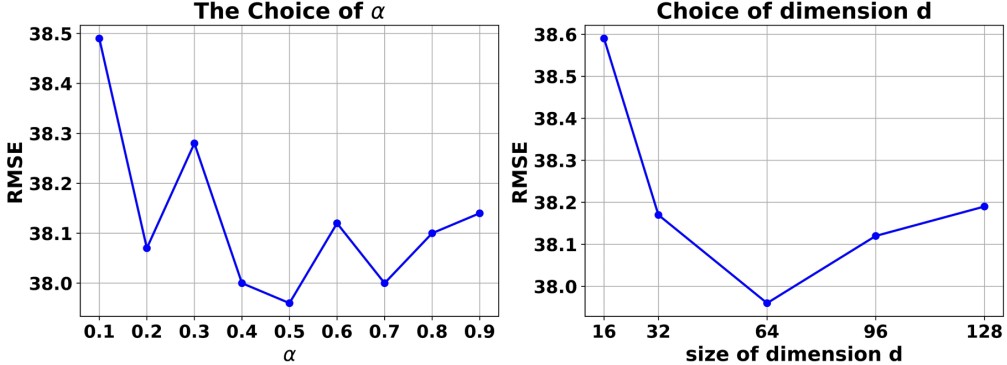

Figure 13: Parameter sensitivity of the proposed SLPF on the average horizon of PEMS08 with weighted selection and $m' = 150$.

Extending the comparison between SLPF and No Rank-based Node Embedding (i.e., SLPF without rank) in Table 3, we present additional RMSE results in Fig. 12 with a varying number $m'$ of unsensed locations from 50 to 150. It shows that the performance gap between SLPF and its variant without rank increases as the value of $m'$ increases. This demonstrates that rank-based node embedding is more beneficial when the number of unsensed locations is high.

### 5.10 Parameter Sensitivity

We study the parameter sensitivity of our model SLPF with two experiments on the average RMSE over 8 hours of traffic forecast on dataset PEMS08 with $m' = 150$. One parameter is $\alpha$, the ratio factor of the two parts in the aggregation step; the other is the embedding dimension $d$. From Fig. 13, the RMSE at $\alpha = 0.5$ is slightly better than the RMSEs at other $\alpha$ values. The RMSE at $d = 64$ is slightly better than the RMSEs at other $d$ values. The reason is that when the dimension $d$ is smaller than 64, SLPF may not fully capture the complex spatio-temporal patterns; when $d$ is larger than 64, SLPF may suffer from overfitting.

## 6 Conclusion

This paper proposes a novel long-term partial sensing forecast model (SLPF). It has a multi-step training/inference structure that enables progressive refinement of model parameters. It introduces a new rank-based node embedding to handle the impact of traffic noises. It uses the node embedding enhanced spatial transfer matrix to enhance data representation across varying phases by shared parameters.

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
