# OpenReview forum: "Spatio-temporal Partial Sensing Forecast of Long-term Traffic"
_TMLR — Accepted by TMLR_

### Review · Reviewer_7cV9 · 2025-06-05

**Summary Of Contributions:**

The paper studies an important problem of long-term traffic forecasting. A spatio-temporal partial sensing forecast model is proposed. Experiments show the effectiveness of the proposed method.

**Audience:**

Yes

**Claims And Evidence:**

Yes

**Requested Changes:**

Please see the weaknesses.

**Strengths And Weaknesses:**

The paper studies an important problem of traffic prediction. The authors propose a spatio-temporal long-term partial sensing model with extensive experiments. Generally, the paper is well-written and easy to follow. The proposed method is sound. However, it is suggested to improve the paper as follows.
1. More recent baselines are required as follows.

[1]. ST-LLM+: Graph Enhanced Spatio-Temporal Large Language Models for Traffic Prediction, TKDE 2025.

[2]. Spatial-Temporal Large Language Model for Traffic Prediction, MDM 2024.

2. It is suggested to compare the efficiency of the proposed method and baselines. In addition, theoretical computational analysis, i.e., time and space complexities, is required.
3. It would be better to include a case study to intuitively show the effectiveness of the proposed method.
4. More recent related work is required.

[3]. MBA-STNet: Bayes-enhanced discriminative multi-task learning for flow prediction, TKDE 2023.

[4]. A unified replay-based continuous learning framework for spatio-temporal prediction on streaming data, ICDE 2024.

[5]. TimeCMA: Towards LLM-Empowered Multivariate Time Series Forecasting via Cross-Modality Alignment, AAAI 2025.

[6]. Towards Cross-Modality Modeling for Time Series Analytics: A Survey in the LLM Era, IJCAI 2025.

---

> ### Author Response · Authors · 2025-07-14
>
> **Reviewer 4**
>
> **Response to Comment 1:**
> We update the experiment table with these two methods, STLLM [2] and STLLM+ [1], in Table 2. Our model still outperforms all the baselines.
>
> **Response to Comment 2:**
> We add complexity analysis in Section 4.6 to address this comment.
>
> **Parameter complexity:**
> Let $n$ be the number of locations (and ranks), $d$ the embedding dimension size, $D = 5d$ the aggregated dimension size, and $\alpha$ the number of MLP layers. The parameter complexity is:
>
> $$
> \mathcal{O}\left( (2n + N^{\text{dow}} + N^{\text{tod}}) \cdot d + n^2 + \alpha \cdot D^2 \right)
> $$
>
> **Time complexity:**
> SLPF has three stages: adaptation, forecast, and aggregation. The dominant terms are:
>
> - Adaptation stage:
>
> $$
> \mathcal{O}_{\text{adp}} = m \cdot l \cdot D + m \cdot D^2 + m \cdot m' \cdot D + m' \cdot D \cdot l
> $$
>
> - Forecast stage:
>
> $$
> \mathcal{O}_{\text{fore}} = n \cdot l \cdot D + n \cdot D^2 + n \cdot m \cdot D + m' \cdot D \cdot l'
> $$
>
> - Aggregation stage:
>
> $$
> \mathcal{O}_{\text{agg}} = (n \cdot l + m \cdot l') \cdot D + (n + m) \cdot D^2 + (n + m) \cdot m' \cdot D + m' \cdot D \cdot l'
> $$
>
> The experimental results in Figure 8 demonstrate empirically that SLPF achieves good efficiency in training time and superior accuracy-efficiency tradeoff with respect to the baselines. This paper proposes two models: LPF achieves high efficiency, while SLPF achieves the best accuracy but less time efficiency due to its multi-module design. Together they provide choices for practical applications.
>
> **Response to Comment 3:**
> Rank embedding represents an additional input, akin to normalized, discrete flow rates, to the model. To address this comment, we enhance the interpretability of the rank embedding mechanism through a series of steps from individual case studies to average performance statistics to hypothesis on the underlying reason to finally experimental verification of the hypothesis.
>
> To begin with, we add case studies in Section 5.4 to visualize the impact of rank embedding. Figure 4 shows the actual flow rates (blue), the forecast rates of SLPF (red), and the forecast rate of SLPF without rank embedding (gray) at Location 25 and Location 117 in the PEMS08 dataset. It clearly demonstrates that rank embedding allows the red line of SLPF to track the underlying pattern by smoothing out the noise (traffic fluctuations) of the actual rate curve, outperforming the gray line of SLPF without rank embedding.
>
> While individual cases are obviously not conclusive, our ablation study in Table 3 already shows the statistical results in MAE, RMSE, and MAPE, which confirm that SLPF (with rank embedding) outperforms its variant without rank embedding in long-term forecast.
>
> But why? We hypothesize in Section 3.2 that node ranks, which are essentially the normalized, discrete flow rates at the nodes, are more stable input to the model than the actual rates, which fluctuate. Hence, rank embedding is more resistant against noise in traffic, and thus helps capture the underlying patterns better. Section 5.4 then continues with experiments (Figure 5,6,7) to verify our hypothesis. The results confirm that rank embedding indeed handles noise better, thanks to its relative stability compared with the actual flow rates.
>
> **Response to Comment 4:**
> We update the related work to include the papers ([3],[4],[5],[6]), see the blue content in Section 1.

---

### Review · Reviewer_PP6v · 2025-06-10

**Summary Of Contributions:**

This paper introduces and tackles the problem of "long-term partial sensing traffic forecast", which aims to predict future traffic flow for many time steps ahead at locations without sensors, using only recent data from a sparse set of sensored locations.  The authors propose two models: LPF, and its more advanced version, SLPF. The key contributions include a rank-based node embedding designed to mitigate the impact of high-frequency noise in traffic data , a "node embedding enhanced spatial transfer matrix" to capture correlations between sensed and unsensed locations , and a multi-step training process for SLPF that leverages additional data available only during training (e.g., from temporary sensors) to progressively refine the model.  Extensive experiments on five real-world datasets show that the proposed models, particularly SLPF, outperform a wide array of adapted baselines in terms of accuracy and offer a good trade-off between accuracy and training efficiency

**Audience:**

Yes

**Claims And Evidence:**

Yes

**Requested Changes:**

1. The description of the experiment in Section 3.1 appears to contradict the legend in Figure 2. For instance, the text states, "we use GinAR to perform the traditional short-term full sensing traffic forecast... The experimental results are presented as the blue line in Fig. 2", but the figure legend labels the blue line as "GinAR short-term partial-sensing".
2. The high-level description of the three-step training in Section 4.2 could be clearer. The paper states that "Each step leverages the module parameters derived from its previous step(s) ...". It is unclear how this is done.
3. For the "No Spatial Transfer Matrix" ablation variant, the paper mentions using "an MLP to map the input sensor dimensions to the desired output dimensions". It would be helpful to specify the structure of this MLP e.g., a single linear layer?
4. typo? In sec 5.8 "Fig. 10 compares our method with the baselines..."

**Strengths And Weaknesses:**

+  The paper defines a practical and challenging problem. Most prior work either assumes full sensor coverage or focuses only on short-term predictions.  Addressing long-term forecasting with partial sensing is a valuable and timely contribution, given the high cost of deploying and maintaining permanent sensors everywhere.
+ The three-step training process for the SLPF model is a smart way to utilize all available data during the training phase, where ground truth for unsensed locations might be temporarily available.
+ The experimental evaluation is comprehensive. The authors test on five standard datasets , compare against a large number of appropriately adapted baselines , and provide detailed ablation studies to validate each component of their model.

- The SLPF model's superior performance hinges on a multi-step training process that requires ground truth data for the unsensed locations during training.  The authors suggest using temporary mobile sensors for this purpose.  While they argue this is cheaper than permanent installation, this assumption represents a significant logistical and financial hurdle for real-world deployment.
- The specific formulation of the "node embedding enhanced spatial transfer matrix" in Equation 3 feels somewhat ad-hoc. While the ablation study shows it works better than simpler alternatives, the theoretical or intuitive justification for this exact structure is not explored.

---

> ### Author Response · Authors · 2025-07-14
>
> **Response to Comment 1:**
> Thanks for pointing it out. We have corrected Figure 2 and the paragraph (in Section 3.1, blue paragraph) to make them consistent.
>
> **Response to Comment 2:**
> Thanks for pointing this out. This sentence is indeed confusing. Our complete model consists of three stacked modules. The three modules — dynamic adaptation, long-term forecasting, and aggregation — are trained sequentially in three steps, where each step uses the previous module(s) and refines their parameters while training its own module.
>
> More specifically:
> - The dynamic adaptation module is first trained in Step 1 to use $X_{M, T}$ to forecast $X_{M', T}$.
> - Then the long-term forecasting module is trained next in Step 2 to use $X_{M, T}$ and $X_{M', T}$ to forecast $X_{M, T'}$, where it "leverages the module parameters in the previous step" because part of its input, $X_{M', T}$, is produced by the dynamic adaptation module from Step 1.
> - Similarly, when we train the aggregation module in Step 3 to use $X_{M, T}$, $X_{M', T}$ and $X_{M, T'}$ to forecast $X_{M', T'}$, part of its input, $X_{M', T}$ and $X_{M, T'}$, are produced by the previous two modules.
>
> The sentence that causes confusion is the last one in the paragraph. After careful evaluation, we believe the paragraph is clear (after some revision) without this sentence. In our judgment, the best way to address this comment is to remove this sentence.
>
> **Response to Comment 3:**
> Thanks for pointing this out. Yes, we have used a single linear layer with a shape of $\mathbb{R}^{M \times M'}$, which is now clarified in the paper.
>
> **Response to Comment 4:**
> Thanks for pointing this out. We correct the typo.

---

> ### Author Response · Authors · 2025-07-14
>
> **Response to Comment 1:**
> Thanks for pointing it out. We have corrected Figure 2 and the paragraph (in Section 3.1, blue paragraph) to make them consistent.
>
> **Response to Comment 2:**
> Thanks for pointing this out. This sentence is indeed confusing. Our complete model consists of three stacked modules. The three modules — dynamic adaptation, long-term forecasting, and aggregation — are trained sequentially in three steps, where each step uses the previous module(s) and refines their parameters while training its own module.
>
> More specifically:
> - The dynamic adaptation module is first trained in Step 1 to use $X_{M, T}$ to forecast $X_{M', T}$.
> - Then the long-term forecasting module is trained next in Step 2 to use $X_{M, T}$ and $X_{M', T}$ to forecast $X_{M, T'}$, where it "leverages the module parameters in the previous step" because part of its input, $X_{M', T}$, is produced by the dynamic adaptation module from Step 1.
> - Similarly, when we train the aggregation module in Step 3 to use $X_{M, T}$, $X_{M', T}$ and $X_{M, T'}$ to forecast $X_{M', T'}$, part of its input, $X_{M', T}$ and $X_{M, T'}$, are produced by the previous two modules.
>
> The sentence that causes confusion is the last one in the paragraph. After careful evaluation, we believe the paragraph is clear (after some revision) without this sentence. In our judgment, the best way to address this comment is to remove this sentence.
>
> **Response to Comment 3:**
> Thanks for pointing this out. Yes, we have used a single linear layer with a shape of $\mathbb{R}^{M \times M'}$, which is now clarified in the paper.
>
> **Response to Comment 4:**
> Thanks for pointing this out. We correct the typo.

---

### Review · Reviewer_Cwax · 2025-06-28

**Summary Of Contributions:**

This paper proposes a **Spatio-temporal Long-term Partial Sensing Forecast (SLPF)** model for long-term traffic flow forecasting. The main contributions include: a ranking-based node embedding technique is proposed. Combining node embedding and ranking-based embedding, the model's knowledge transfer ability from sensor-based areas to sensor-free areas is enhanced, effectively solving the problem of spatial distribution offset. Using three steps of dynamic adaptation, long-term prediction and aggregation, the model parameters are gradually refined, the training data is fully utilized, and the prediction accuracy is improved.

**Audience:**

Yes

**Claims And Evidence:**

Yes

**Requested Changes:**

1. **Model complexity optimization**: It is recommended to explore how to reduce the computational complexity while maintaining model performance in subsequent work, such as through model compression or more efficient network structure design.

2. **Enhanced interpretability**: It is necessary to further analyze the specific impact mechanism of ranking embedding on spatial transfer matrix and long-term prediction, and increase the transparency of the model.

3. **Expand the scope of experiments**: It is recommended to conduct experiments on more diverse traffic datasets (such as LargeST Datasets) to verify the universality of the model.

4. **Discussion and comparison of related work**: Supplement the comparison of competing work and the discussion of related work.

**Strengths And Weaknesses:**

## Strengths

1. **Comprehensive Experiments**: Extensive experiments are conducted on multiple real datasets, including datasets of different scales and geographical locations, verifying the universality and effectiveness of the model.

2. **Good Robustness**: The stability of the model under different noise levels is demonstrated through comparative experiments, proving its potential in practical applications.

---
## Weaknesses

1. **Lack of Related Work**: The motivation of this paper actually comes from the challenge of spatiotemporal OOD generalization, so the author lacks discussion and comparison of some related works. I list some key works for the authors to read. [1] gives an inductive setting for spatiotemporal learning. [2] systematically explains the OOD generalization setting for spatiotemporal learning, [3] systematically explains the continual learning setting for long-term dynamic evolution of spatiotemporal learning, and [4] further systematically summarizes these works and provides another perspective. In addition to these works, some baseline related works in their paper need to be discussed.

[1] Lei, et al. ST-FiT: Inductive Spatial-Temporal Forecasting with Limited Training Data. AAAI, 2025.

[2] Ma, et al. STONE: A Spatio-temporal OOD Learning Framework Kills Both Structural and Temporal Shifts. KDD, 2024.

[3] Chen, et al. Expand and Compress: Exploring Tuning Principles for Continual Spatio-Temporal Graph Forecasting. ICLR, 2025.

[4] Chen, et al. Learning with Calibration: Exploring Test-Time Computing of Spatio-Temporal Forecasting. arXiv, 2025.

2. **High computational complexity**: Multi-step training and multi-module design may lead to high consumption of computing resources, limiting its application in large-scale real-time systems.

3. **Lack of theoretical analysis**: Although ranking embedding is proposed, the explanation of how it specifically affects spatial transfer and long-term prediction is not in-depth enough.

4. **Lack of analysis of long-term stability of the model**: The possibility of parameter drift or performance degradation of the model during long-term operation and the coping strategies are not discussed in detail.

---

> ### Author Response · Authors · 2025-07-14
>
> **Response to Weakness 1 and Comment 1:**
> A great suggestion! We apply structural pruning on MLPs, disabling 30% of neurons based on their L2 norm. Specifically, we reduce the training time on the PEMS08 dataset from 40 minutes to 28 minutes, without noticeable performance drop. As shown in Table 5, SLPF with structural pruning achieves comparable forecasting accuracy across all datasets.
>
> **Response to Weakness 2:**
> We add complexity analysis in Section 4.6 to address this comment.
>
> **Parameter complexity:**
> Let $n$ be the number of locations (and ranks), $d$ the embedding dimension size, $D = 5d$ the aggregated dimension size, and $\alpha$ the number of MLP layers. The parameter complexity is:
>
> $$
> \mathcal{O}\left( (2n + N^{\text{dow}} + N^{\text{tod}}) \cdot d + n^2 + \alpha \cdot D^2 \right)
> $$
>
> **Time complexity:**
> SLPF has three stages: adaptation, forecast, and aggregation. The dominant terms are:
>
> - Adaptation stage:
>
> $$
> \mathcal{O}_{\text{adp}} = m \cdot l \cdot D + m \cdot D^2 + m \cdot m' \cdot D + m' \cdot D \cdot l
> $$
>
> - Forecast stage:
>
> $$
> \mathcal{O}_{\text{fore}} = n \cdot l \cdot D + n \cdot D^2 + n \cdot m \cdot D + m' \cdot D \cdot l'
> $$
>
> - Aggregation stage:
>
> $$
> \mathcal{O}_{\text{agg}} = (n \cdot l + m \cdot l') \cdot D + (n + m) \cdot D^2 + (n + m) \cdot m' \cdot D + m' \cdot D \cdot l'
> $$
>
> The experimental results in Figure 8 demonstrate empirically that SLPF achieves good efficiency in training time and superior accuracy-effiency tradeoff with respect to the baselines. This paper proposes two models: LPF achieves high efficiency, while SLPF achieves the best accuracy, but less time efficiency due to its multi-module design. Together they provide choices for practical applications.
>
> **Response to Weakness 3 and Comment 2:**
> Rank embedding represents an additional input, i.e., normalized, discrete flow rates, to the model. To address this comment, we enhance the interpretability of the rank embedding mechanism through a series of steps from individual case studies to average performance statistics to hypothesis on the underlying reason to finally experimental verification of the hypothesis.
>
> - First, we use case studies to visualize the impact of rank embedding. Figure 4 shows the actual flow rates (blue), the forecast rates of SLPF (red), and the forecast rate of SLPF without rank embedding (gray) at Location 25 and Location 117 in the PEMS08 dataset. It clearly demonstrates that rank embedding allows the red line of SLPF to track the underlying pattern by smoothing out the noise (traffic fluctuations) of the actual rate curve, outperforming the gray line of SLPF without rank embedding.
>
> - Second, while individual cases are obviously not conclusive, our ablation study in Table 3 shows the statistical results in MAE, RMSE, and MAPE, which confirm that SLPF (with rank embedding) outperforms its variant without rank embedding in long-term forecast.
>
> - But why? We hypothesize in Section 3.2 that node ranks, which are essentially the normalized, discrete flow rates at the nodes, are more stable input to the model than the actual rates, which fluctuate. Hence, rank embedding is more resistant against noise in traffic, and thus helps capture the underlying patterns better.
>
> - Finally, we use experiments (Figure 5,6,7) to verify our hypothesis. The results confirm that rank embedding indeed handles noise better, thanks to its relative stability compared with the actual flow rates.
>
> **Response to Comment 3:**
> We add new experiments on two datasets (SD and GBA) from the LargeST Dataset. The results are in Table 6, which shows the proposed model SLPF outperforms the most related work GinAR, demonstrating its superiority on missing-sensor forecast.

---

> > ### Author Response · Authors · 2025-07-14
> >
> > **Response to Comment 4:**
> > Thanks for your comment. We add a new paragraph (in blue) in Section 1 as the supplementary comparison to address your concern. We agree that recent progress in spatio-temporal OOD generalization, including ST-FiT [1], STONE [2], Expand-and-Compress [3], and Learning with Calibration [4], is relevant. However, these problems differ from our setting.
> >
> > SF-FiT focuses on predicting future values for all sensors based on inductive learning from a subset of sensors whose temporal data is available during training. Crucially, it assumes that the target sensors, i.e., those unobserved during training, have no historical data at all, but all sensors are required to be observed during inference to generate predictions. This differs fundamentally from our setting, where we assume that temporary sensors are deployed during training to collect historical data for all locations, but only a subset of sensors remains available during inference. Furthermore, ST-FiT focuses on short-term forecasting, while our work addresses the significantly more challenging problem of long-term forecasting under partial sensing, where issues like error accumulation and noise amplification become substantially more severe.
> >
> > STONE tackles spatio-temporal OOD generalization by addressing spatial shifts and temporal shifts, leveraging techniques such as Fréchet embedding to represent dynamic spatial graphs. However, STONE assumes that the node set dynamically evolves over time (i.e., locations can appear or disappear). In contrast, our setting assumes a static graph topology, where the set of locations is fixed, but only a subset is equipped with sensors during inference.
> >
> > Expand-and-Compress addresses continual learning on spatio-temporal graphs by dynamically expanding the spatial dimension using a learnable node pool, and compressing parameters via low-rank adaptation. While this framework is designed for handling distribution shifts over time, it differs from our setting, which focuses on spatial distribution shifts between sensed and unsensed nodes under a static graph structure without continual evolution.
> >
> > Learning with Calibration addresses test-time distribution shifts by performing online adaptation through a spectral domain calibrator and a flash gradient update mechanism. It leverages label autocorrelation in spatio-temporal data to adjust model outputs during inference. In contrast, our model remains frozen at inference time without any test-time updates. Our primary focus is on handling spatial distribution shifts under partial sensing, which is different from the temporal non-stationarity correction targeted by Learning with Calibration.

---

> > > ### Comment · Reviewer_Cwax · 2025-07-14
> > >
> > > I appreciate the author's hard work during these days. I have read the author's response carefully and am generally satisfied. However, I am still confused about the practicality of the author's current setting, that is, you assume that all sites and data have been seen during training, and some site data are missing during testing. Can this be regarded as an extreme case of spatiotemporal forecasting under noise or missing data? I recommend that the author briefly explain it with an example or a figure example in the introduction so that the author can quickly understand it. In addition, for this problem under noise or missing data, I list some work [1, 2] for your reference.
> > >
> > > ---
> > >
> > > [1] Information Bottleneck-guided MLPs for Robust Spatial-temporal Forecasting. ICML, 2025.
> > >
> > > [2] Graph-based forecasting with missing data through spatiotemporal downsampling. ICML, 2024.

---

### Review · Reviewer_yA4H · 2025-07-02

**Summary Of Contributions:**

This paper addresses the challenging problem of long-term traffic forecasting with partial sensor coverage, where only a subset of locations are equipped with sensors. To tackle issues like unknown data at unsensed locations, spatio-temporal complexity, and noisy traffic patterns, the authors propose a novel model called Spatio-temporal Long-term Partial sensing Forecast. Key innovations include a rank-based embedding to mitigate noise, a spatial transfer matrix to handle distribution shifts between sensed and unsensed locations, and a multi-step training strategy that progressively refines model parameters using all available data. Extensive experiments on real-world traffic datasets show that the proposed model significantly outperforms existing methods in forecasting accuracy.

**Audience:**

Yes

**Broader Impact Concerns:**

None.

**Claims And Evidence:**

Yes

**Requested Changes:**

Comments:

- From Table 3, I do not seem to see that different modules in SLPF have a significant impact on SLPF performance, because their experimental results are relatively close. Could the authors provide more explanation?

- From Figure 4, it seems that SLPF performs worse as the number of m' increases. So the authors need to explain at least two things: 1) how the performance of other baselines changes with the change of m'; 2) this paper considers the problem of missing sensors, but the results do not seem to solve this problem well.

- This paper only uses the Gaussian noise model to verify the robustness of SLPF. I would like to know how SLPF performs under other noise models.

- Considering the complexity of the SLPF structure, it is necessary to analyze the algorithm complexity theoretically.

**Strengths And Weaknesses:**

Strengths:
+ This work presents SLPF, a long-term traffic forecasting model with partial sensor coverage, where only a subset of locations is equipped with sensors.
+ Extensive case studies.

Weaknesses:
- Different noise models need to be considered to further verify the performance of SLPF.
- The time complexity and space complexity analysis of the algorithm need to be supplemented.
- The impact of different numbers of missing sensors on the performance of SLPF needs to be further explored.

---

> ### Author Response · Authors · 2025-07-13
>
> **Response to Comment1:**
> We will clarify in the revision to address the above comment. The ablation study considers 5 variants, each disabling one design component of our final model (SLPF). The goal is not to compare amongst themselves (many of their numbers are close), but to compare each of them with SLPF to show the impact of the disabling component, which is about 1–2% MAE degradation. Such margin is considerable in this line of research (TESTAM [1], GinAR [2]), as seen from the comparison in Table 3. Hence, the study justifies the importance of those design components, which contribute to SLPF in different ways: rank improves robustness against noise, transfer matrix enhances spatial generalization, and temporal embeddings capture periodic patterns, etc. With them together, SLPF outperforms the existing work by larger margins in Table 2, and the gap is much larger when we consider worst-case scenarios in Section 5.7.
>
> **Response to Comment2:**
> We thank the reviewer for this valuable comment. In Figure 10, we conduct the experiments with varying unsensed locations ($m'$) and add new experimental results (Figure 7) to address it. It is true that SLPF's performance decreases as the number $m'$ of unsensed locations increases in Figure 7. That is, however, expected for SLPF and all baselines, because there is less sensed information available to make forecasts. Yet, SLPF consistently outperforms all baselines across all $m'$ levels and the performance gap increases with more missing sensors, suggesting its effectiveness in addressing the missing sensor problem with respect to the state of the art (the baselines).
>
> **Response to Comment3:**
> To extend the noise-impact study in Figure 4 beyond Gaussian noise, we add two more commonly used noise distributions, using a shared scale parameter $\gamma$ for magnitude control:
>
> - **Uniform noise** $\mathcal{U}(-\gamma, \gamma)$:
>   $$
>   p(x) = \begin{cases}
>     \frac{1}{2\gamma}, & \text{if } x \in [-\gamma, \gamma] \\
>     0, & \text{otherwise}
>   \end{cases}
>   $$
>
> - **Laplace noise** $\text{Laplace}(0, \gamma)$:
>   $$
>   p(x) = \frac{1}{2\gamma} \exp\left( -\frac{|x|}{\gamma} \right)
>   $$
>
> The new experimental results are presented in Figure 5 and 6 for uniform noise and Laplace noise, respectively. The results agree with Figure 4 (Gaussian noise) that rank improves SLPF's robustness against noise.
>
> **Response to Comment4:**
> We add complexity analysis to address this comment.
>
> **Parameter complexity:**
> Let $n$ be the number of locations (and ranks), $d$ the embedding dimension size, $D = 5d$ the aggregated dimension size, and $\alpha$ the number of MLP layers. The parameter complexity is:
> $$
> \mathcal{O}\left( (2n + N^{\text{dow}} + N^{\text{tod}}) \cdot d + n^2 + \alpha \cdot D^2 \right)
> $$
>
> **Time complexity:**
> SLPF has three stages: adaptation, forecast, and aggregation. The dominant terms are:
>
> - Adaptation stage:
>   $$
>   \mathcal{O}_{\text{adp}} = m \cdot l \cdot D + m \cdot D^2 + m \cdot m' \cdot D + m' \cdot D \cdot l
>   $$
>
> - Forecast stage:
>   $$
>   \mathcal{O}_{\text{fore}} = n \cdot l \cdot D + n \cdot D^2 + n \cdot m \cdot D + m' \cdot D \cdot l'
>   $$
>
> - Aggregation stage:
>   $$
>   \mathcal{O}_{\text{agg}} = (n \cdot l + m \cdot l') \cdot D + (n + m) \cdot D^2 + (n + m) \cdot m' \cdot D + m' \cdot D \cdot l'
>   $$
>
> The experimental results in Figure 8 demonstrate empirically that SLPF achieves high efficiency in training time and notably superior accuracy-efficiency tradeoff with respect to the baselines.
>
> [1]Lee, Hyunwook, and Sungahn Ko. "TESTAM: A Time-Enhanced Spatio-Temporal Attention Model with Mixture of Experts." The Twelfth International Conference on Learning Representations.
> [2]Yu, Chengqing, et al. "Ginar: An end-to-end multivariate time series forecasting model suitable for variable missing." Proceedings of the 30th ACM SIGKDD conference on knowledge discovery and data mining. 2024.

---

> ### Author Response · Authors · 2025-07-14
> **Revised Comment (Please Ignore the Previous Comment)**
>
> **Response to Comment1:**
> We will clarify in the revision to address the above comment. The ablation study considers 5 variants, each disabling one design component of our final model (SLPF). The goal is not to compare amongst themselves (many of their numbers are close), but to compare each of them with SLPF to show the impact of the disabling component, which is about 1–2% MAE degradation. Such margin is considerable in this line of research, as seen from the comparison in Table 3. Hence, the study justifies the importance of those design components, which contribute to SLPF in different ways: rank improves robustness against noise, transfer matrix enhances spatial generalization, and temporal embeddings capture periodic patterns, etc. With them together, SLPF outperforms the existing work by larger margins in Table 2, and the gap is much larger when we consider worst-case scenarios in Section 5.6.
>
> **Response to Comment2:**
>  It is true that SLPF’s performance decreases as the number $m'$ of unsensed locations increase in Figure 5 (previously Figure 4). That is expected for SLPF and all baselines, because there is less sensed information available to make forecast. We respectfully point out that we have experimental results on the performance of other baselines with the change of $m'$ in Fig. 10 of Section 5.7, where the horizontal axis is $m'$, i.e., the number of unsensed locations. SLPF consistently outperforms all baselines with different $m'$ values, suggesting its effectiveness in addressing the missing sensor problem, with respect to the state of the art (the baselines).
>
> **Response to Comment3:**
> To extend the noise-impact study in Figure 5 beyond Gaussian noise, we add two more commonly used noise distributions, using a shared scale parameter $\gamma$ for magnitude control:
>
> - **Uniform noise** $\mathcal{U}(-\gamma, \gamma)$:
>   $$
>   p(x) = \begin{cases}
>     \frac{1}{2\gamma}, & \text{if } x \in [-\gamma, \gamma] \\
>     0, & \text{otherwise}
>   \end{cases}
>   $$
>
> - **Laplace noise** $\text{Laplace}(0, \gamma)$:
>   $$
>   p(x) = \frac{1}{2\gamma} \exp\left( -\frac{|x|}{\gamma} \right)
>   $$
>
> The new experimental results are presented in Figure 6 and 7 for uniform noise and Laplace noise, respectively. The results agree with Figure 5 (Gaussian noise) that rank improves SLPF's robustness against noise.
>
> **Response to Comment4:**
> We add complexity analysis to address this comment.
>
> **Parameter complexity:**
> Let $n$ be the number of locations (and ranks), $d$ the embedding dimension size, $D = 5d$ the aggregated dimension size, and $\alpha$ the number of MLP layers. The parameter complexity is:
> $$
> \mathcal{O}\left( (2n + N^{\text{dow}} + N^{\text{tod}}) \cdot d + n^2 + \alpha \cdot D^2 \right)
> $$
>
> **Time complexity:**
> SLPF has three stages: adaptation, forecast, and aggregation. The dominant terms are:
>
> - Adaptation stage:
>   $$
>   \mathcal{O}_{\text{adp}} = m \cdot l \cdot D + m \cdot D^2 + m \cdot m' \cdot D + m' \cdot D \cdot l
>   $$
>
> - Forecast stage:
>   $$
>   \mathcal{O}_{\text{fore}} = n \cdot l \cdot D + n \cdot D^2 + n \cdot m \cdot D + m' \cdot D \cdot l'
>   $$
>
> - Aggregation stage:
>   $$
>   \mathcal{O}_{\text{agg}} = (n \cdot l + m \cdot l') \cdot D + (n + m) \cdot D^2 + (n + m) \cdot m' \cdot D + m' \cdot D \cdot l'
>   $$
>
> The experimental results in Figure 8 demonstrate empirically that SLPF achieves high efficiency in training time and notably superior accuracy-efficiency tradeoff with respect to the baselines.

---

> > ### Comment · Reviewer_yA4H · 2025-07-16
> > **Thanks for Authors' Rebuttal**
> >
> > I appreciate the clarification from the authors, which has addressed most of my concerns. I will seriously reconsider my recommendation and update this in the final decision.

---

### Decision · Action_Editor_tY1E · 2025-08-01

**Recommendation:** Accept as is

**Audience:**

Yes

**Audience Explanation:**

The work tackles a challenge problem, long-term forecasting with partial sensor coverage, with extensive case studies. The approach is reasonable, and the results are supportive of the approach.

**Claims And Evidence:**

Yes

**Claims Explanation:**

The reviewers found the experiments convincing and extensive in general. While they raised concerns about the clarity of some parts of the paper, the rebuttal addressed those well. (Reviewer PP6v submitted their recommendation before the authors' rebuttal. AE carefully read the rebuttal to this reviewer's comments and found it convincing.)